# GMM-based VAE model with Normalizing Flow for effective stochastic segmentation

**Conghui Li**
School of IT
Monash University Malaysia
Bandar Sunway, Selangor, 47500, Malaysia
`conghui.li@monash.edu`

**Chern Hong Lim**[*]
School of IT
Monash University Malaysia
Bandar Sunway, Selangor, 47500, Malaysia
`Lim.ChernHong@monash.edu`

**Xin Wang**[*]
School of Engineering
Monash University Malaysia
Bandar Sunway, Selangor, 47500, Malaysia
`Wang.Xin@monash.edu`

## Abstract

While deep neural networks possess the capability to perform semantic segmentation, producing a single deterministic output limits reliability in safety-critical applications caused by uncertainty and annotation variability. To address this, stochastic segmentation models using Conditional Variational Autoencoders (CVAE), Bayesian networks, and diffusion have been explored. However, existing approaches suffer from limited latent expressiveness and interpretability. Furthermore, our experiments showed that models like Probabilistic U-Net rely excessively on high latent variance, leading to posterior collapse. This work propose a novel framework by integrating Gaussian Mixture Model (GMM) with Normalizing Flow (NF) in CVAE for stochastic segmentation. GMM structures the latent space into meaningful semantic clusters, while NF captures feature deformations with quantified uncertainty. Our method stabilizes latent distributions through constrained variance and mean ranges. Experiments on LIDC, Crack500, and Cityscapes datasets show that our approach outperformed state-of-the-art in curvilinear structure and medical image segmentation.

## 1 Introduction

In recent years, deep neural networks have made remarkable progress in semantic segmentation tasks. The main goal of most methods lies in generating a single segmentation result that is highly consistent with the content of the image. However, this deterministic segmentation has limitations in safety-critical areas such as medical diagnosis, industrial testing, or autonomous driving. This is due to the images in such applications are often accompanied by inherent uncertainty, leading to the presence of multiple reasonable yet contradictory manual annotations. [41, 28]. In this context, providing a single deterministic segmentation result may not be able to fully express the possible true semantic distribution of the image, thus limiting the reliability and practicability of the model in these critical tasks.

Currently, many studies have proposed probabilistic approches for sampling from output distributions, including models based on Conditional Variational Autoencoders (CVAE) [25, 24], Normalizing

---

[*]Corresponding author.

Flows [35], Bayesian neural network [16, 14, 8] and Diffusion [32, 41]. Most of the study recognized the Gaussian distribution's limitation in capturing true feature distributions. Therefore, in recent years, researchers have been dedicated to enhancing the ability of variational autoencoders (VAE) in complex data modeling by replacing the traditional simple Gaussian distribution with a more complex distribution to improve the expressive power of the distribution and capture more complex semantic information [36, 33, 37, 29]. An alternative is to use the diffusion model to capture the data distribution more flexibly through a multi-step denoising process [32, 41]. However, the drawback of diffusion is the unexplainability of distribution, which is crucial for security-sensitive tasks. Therefore, we advocate returning to the VAE framework and further enhancing the latent space modeling mechanism to make up for the deficiencies of the original Gaussian hypothesis.

We performed deeper analysis to prove the ineffective of Gaussian distribution by experimenting the existing Probabilistic segmentation model, Probabilistic UNet [25]. The results show that the variance of its latent variables was significantly higher than the mean after training, reaching a 500-fold gap in the curvilinear structure segmentation task. This suggests that the latent spaces fail to learn compact semantic representations and relying on high variance to cover a wide range of possibilities, leading models to favor guesswork over deterministic predictions. To examine the effectiveness of Gaussian latent variables, we modified the Probabilistic U-Net by replacing the latent samples with purely random vectors. The results in Figure 1 show that the model's performance does not degrade significantly which indicates that the latent variables contribute limited semantic information.

To address the aforementioned limitations, we propose a novel framework that integrates Gaussian Mixture Model (GMM) and Normalizing Flow (NF) on a Conditional Variational Autoencoder (CVAE). The GMM can define semantic clusters in the latent space more effectively, where each Gaussian component captures a distinct mode of variation corresponding to meaningful visual concepts or feature patterns. In detail, each Gaussian component learns distinct features through a parameter competition mechanism, mitigating the prediction ambiguity caused by a single Gaussian prior distribution. Based on the latent space provided by GMM, NF learns the reasonable deformation of image features, and its Jacobian determinant directly quantifies the local uncertainty. Instead of only let the posterior distribution fitting the data, we apply this combined model to both



Figure 1: Qualitative comparison of using random number and gaussian distribution for latent variable in Prob. Unet [25]. 1-2 is the result from gaussian distribution, 3-4 is the result from random number

the prior and posterior distributions, and introduce constraints on the range of means and variances to stabilize NF's training gradients.

Worth to mention that We bring this approach to curvilinear structures for the first time. Experimental results demonstrate superior performance in wide range of applications such as medical image segmentation, infrastructure crack detection, and city scene segmentation using LIDC-IDRI, Crack500 dataset, and cityscape dataset respectively. In infrastructure crack detection tasks (using Crack500 dataset), our method achieves a better balance between recall and precision, improving the F1-score by 11.0% compared to the standard CVAE-based stochastic segmentation. Furthermore, evaluations on the multi-label segmentation datasets, LIDC-IDRI and Cityscapes datasets show that our approach consistently outperformed the existing baseline models.

In summary, the contributions of this paper are as follows:

1. We propose a combination of GMM through a Multiple-Input, Multiple-Output (MIMO) mechanism to construct a structured latent space in CVAE. GMM provides multiple potential semantic clusters to reduce the transformation complexity of NF and thereby improve performance.

2. We introduce NF into both the prior and posterior distributions to addresses the semantic ambiguity inherent in traditional VAEs caused by overly simplistic priors.

3. We are the first to apply a stochastic segmentation model to curvilinear structure segmentation and achieve state-of-the-art performance compared to the best existing baselines.

## 2 Related work

The goal of probabilistic semantic segmentation is to capture both aleatoric uncertainty and the inherent ambiguity present in segmentation labels. A variety of methods have been proposed to address this. Early approaches employed Bayesian neural networks, in which model parameters were treated as probability distributions and sampled during inference [8, 42, 30, 1, 16] . While this enabled uncertainty modeling, it significantly increased inference overhead and model complexity due to repeated sampling and the need to maintain distributions for each parameter.

To address these limitations, researchers later adopted Conditional Variational Autoencoder (CVAE) frameworks for semantic segmentation. Initial CVAE models were typically built on U-Net architectures, where a latent variable was concatenated with the last feature layer to introduce randomness into the segmentation process [25]. Subsequent work extended this by incorporating hierarchical latent structures, injecting randomness at multiple levels of the encoder-decoder pipeline of U-Net to enhance expressiveness [24, 3].

However, a major limitation persisted: the use of simple Gaussian distributions in both the prior and posterior, which restricted the model's ability to capture complex, multi-modal semantic representations. To overcome this, researchers engaged more expressive probabilistic structures [36, 7, 23, 33]. For example, cFlow applies normalizing flow (NF) to the posterior to enhance flexibility and mitigate posterior collapse. Despite these improvements, some studies argue that such models still underperform in accuracy-sensitive applications due to their limited prior modeling capabilities [35].

To address this, Zhang et al. proposed the Joint Probabilistic U-Net (JProb. Unet), which employs a Reversible multi-layer perceptron (MLP) to simultaneously transform both the prior and posterior distributions into more complex forms [45]. However, the transformation capacity of Reversible MLPs is limited compared to conventional flow-based methods, often requiring deep stacking to approximate complex distributions. Additionally, its grouped variable structure lacks the ability to model global dependencies effectively.

In this work, we propose a stochastic segmentation framework that combines a Gaussian Mixture Model (GMM) with a Normalizing Flow (NF) to construct a flexible and structured latent space. The GMM enables explicit modeling of multi-modal semantic priors, while the NF applies nonlinear transformations to each Gaussian component, significantly enhancing the capacity of both the prior and posterior distributions. Compared with traditional CVAE approaches that assume a single Gaussian, our method can effectively mitigates issues such as posterior collapse and semantic ambiguity. It also demonstrates superior modeling capability for fine-grained structures such as cracks. Experimental results show that our approach achieves notable improvements on tasks involving curvilinear structure segmentation, medical image uncertainty modeling, and multi-label scene understanding—highlighting its strong potential for high-precision and interpretable stochastic segmentation applications.

## 3 Methodology

### 3.1 Gaussian mixture distribution

In most existing VAE based image segmentation models, the latent variable is usually modeled as a unimodal Gaussian distribution $N(\mu, \sigma)$. However, our experiments in Table 1 show that this simple distribution hypothesis is difficult to effectively capture complex and structured semantic information in images. To address these limitations, we introduce the Gaussian Mixture Model (GMM) as the base distribution for modeling latent variables to improve the initialization (Figure 2).

GMM is a multimodal probabilistic model defined as a weighted sum of multiple Gaussian components, enabling the representation of diverse semantic clusters. Compared to a single Gaussian, GMM offers greater representational capacity, allowing the latent space to explicitly encode distinct semantic regions in the image. Through a parameter competition mechanism, each Gaussian component is encouraged to specialize in specific semantic attributes, thereby enhancing both the interpretability of the latent space and the clarity of the resulting segmentations.

In this work, we adopt MIMO mechanism to generate the parameters of a Gaussian Mixture Model. Each input header receives the same image input but is designed to extract different semantic perspectives or preferences from that input. A shared backbone network encodes the global contextual

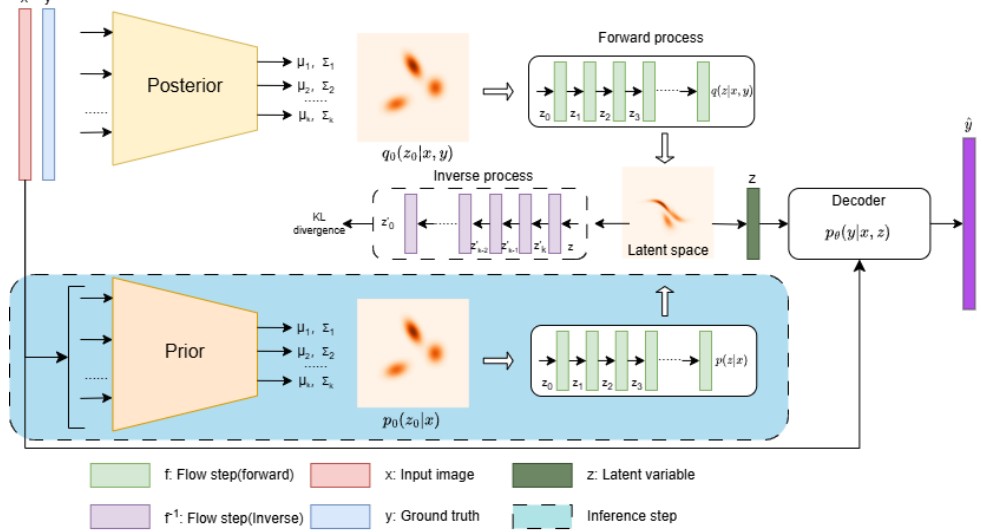

Figure 2: Overview of the GMM-based VAE with Normalizing Flow. During training, the input image $x$ and label $y$ are concatenated and duplicated as inputs to the posterior network, which outputs the mean and covariance for each GMM component to construct the latent space $q_0(z_0|x, y)$. A multi-layer NF transforms this space into a more expressive latent distribution $q(z|x, y)$. KL divergence is computed using the inverse NF in the prior and the forward NF in the posterior to align their distributions. During inference, the prior network generates the latent space $p(z|x)$ via GMM and NF, from which a latent variable $z$ is sampled and fused into the feature maps of Decoder $p_\theta(y|x, z)$ to introduce stochastic segmentation output $\hat{y}$.

information of the image, serving as a common feature foundation for all branches. Each output head then generates the parameters of the Gaussian components, producing means and covariances as a set:

$$\{(\mu_1, \Sigma_1), (\mu_2, \Sigma_2), \ldots, (\mu_o, \Sigma_o)\} \tag{1}$$

Where $o$ represent the number of output in MIMO framework and the number of GMM component. Additionally, a separate head predicts the component weights $\pi$, forming the complete mixture model. The resulting Gaussian Mixture distribution is defined as:

$$p_0(z) = \sum_{o=1}^{O} \pi_o \cdot \mathcal{N}(z \mid \mu_o, \Sigma_o) \tag{2}$$

Among them, a single Gaussian distribution term is defined as:

$$\mathcal{N}(z \mid \mu_o, \Sigma_o) = \frac{1}{(2\pi)^{d/2}|\Sigma_o|^{1/2}} \exp\left(-\frac{1}{2}(z - \mu_o)^T \Sigma_o^{-1}(z - \mu_o)\right) \tag{3}$$

### 3.2 Normalizing flow

NF enhances probabilistic modeling by transforming a simple base distribution into a more expressive and flexible one through a sequence of learnable, invertible transformations.

Taking the prior distribution in a CVAE as an example, assume the initial latent variable is sampled from a base distribution: $z_0 \sim p_0(x)$. A normalizing flow function is typically constructed as a composition of multiple transformations $f_k$, where the subscript $k$ denotes the number of flow layers. The final transformed latent variable is given by:

$$z = f_K \circ f_{K-1} \circ \cdots \circ f_1(z_0) \tag{4}$$

In stochastic segmentation task, aligning the learned latent distribution with the true data distribution enhances the expressiveness of the posterior and alleviates posterior collapse, a phenomenon where the approximate posterior $q(z|x,y)$ degenerates the prior $p(z|x)$, causing the $KL divergence = 0$. This lead samples generated by VAE lack diversity, and the semantic features extracted by the latent variable z are not obvious. Secondly, incorporating Normalizing Flow (NF) into the prior distribution enables the sampling of latent variables that encode richer semantic information. By learning a more flexible and data-adaptive prior, NF encourages the latent variables to align with meaningful structures in the input space instead of providing more noise for randomness, thus improving both the interpretability and effectiveness of the generative process.

In conventional NF frameworks, the distribution to be transformed is referred to as the base distribution, which is typically fixed—for example, a standard normal distribution. However, the parameters of base distribution evolve during training in CVAE framework. This feature introduces challenges for NF application in CVAE. Specifically, in the early stages of training, before the flow is fully trained, the base distribution tends to exhibit high variance to cover the uncertain region due to its limited expressive capacity. As a result, the sampled latent variables are highly dispersed, making it difficult for the NF to learn effective and stable transformations. We applied segmented $\beta-$annealing algorithm to mitigate this issue, but it still happen by chance. Therefore, we impose constraints on the mean and variance of each Gaussian component in the GMM base distribution, enforcing the following range $\frac{\sigma}{20} < \mu < \sigma * 20$. This constraint ensures the latent variables remain within a reasonable range, promoting smoother and more stable NF transformations throughout training. In our implementation, we utilize a three-layer Neural Spline Flows (NSF) [13] to transform the GMM-based base distribution, leveraging the NSF's capability to model complex, non-linear mappings in the latent space. Different with the simple transformation, NSF use the monotonic rational-quadratic splines as the elementwise transformation, which offers much greater flexibility than affine transforms. Specific explanation of NSF can be found at appendix A.1.

To recap, we choose GMM + NF over Gaussian + NF due to the limitations of a single Gaussian, which is unimodal and overly simplistic. While NFs are theoretically capable of mapping such a base distribution into a complex, multimodal target, doing so requires deep and intricate transformations that can hinder convergence, especially during early training. In contrast, a GMM naturally introduces a multimodal prior, allowing the NF to focus on localized refinements such as fine-tuning, stretching, or twisting individual modes rather than learning global semantic structure from scratch.

### 3.3 Evidence lower bound

The Evidence Lower Bound (ELBO) is the objective function used to train Variational Autoencoder (VAE). It provides a lower bound on the true data log-likelihood and balances Reconstruction loss and Regularization. In this task, Reconstruction loss is referred to Dice, and Regularization is referred as KL divergence.

**Log probabilities**: In this method, we used normalizing flow. The probability density function of the posterior is:

$$q(z|x,y) = q_0(z_0|x,y) \cdot \prod_{k=1}^{K} \left| \det \left( \frac{\partial f_k}{\partial h_{k-1}} \right) \right|^{-1} \tag{5}$$

For the flow log probability, we usually need to add a Jacobian determinant on the basis of the log probability:

$$\log q(z|x,y) = \log q_0(z_0|x,y) - \sum_{k=1}^{K} \log \left| \det \left( \frac{\partial f_k}{\partial h_{k-1}} \right) \right| \tag{6}$$

Where $q$ is the posterior probability density function and $q_0$ represents the base distribution before the flow operation, which in this study is the probability density function of GMM. $det(\cdot)$ stands for Jacobian determinant.

The significance of the Jacobian determinant in NF lies in ensuring density consistency between the base distribution and the transformed distribution. The term $\log q_0(z_0|x, y)$ corresponds to the log-probability of the sample under the base distribution, while the second term accounts for the log-volume change introduced by the flow transformation. This correction ensures that probability mass is preserved and that the final distribution accurately reflects the transformation applied to the base.

In the posterior, since the flow is constructed as a sequence of transformations, we define it as a chain of $K$ invertible functions. Let:

$$z_1 = f_1(z_0), \quad z_2 = f_2(z_1), \quad \ldots, \quad z_K = f_K(z_{K-1}) = z \tag{7}$$

This forward transformation is used in the posterior flow. In the prior, $f_K^{-1}$ represents the inverse function and $z'$ represent the inversed latent variable, the process is defined in the same way. The total log-volume change of the transformation is calculated by summing the log-determinants of the Jacobians at each layer. Since each transformation $f_k$ is invertible, we can compute the probability density of the transformed latent variable $z$ using the change-of-variables formula:

$$p(z|x) = p_0(z_0'|x) \cdot \prod_{k=1}^{K} \left| \det \left( \frac{\partial f_k^{-1}}{\partial z_{k-1}'} \right) \right| \tag{8}$$

Taking the logarithm yields the log-probability:

$$\log p(z|x) = \log p_0(z_0'|x) + \sum_{k=1}^{K} \log \left| \det \left( \frac{\partial f_k^{-1}}{\partial z_{k-1}'} \right) \right| \tag{9}$$

**KL divergence**: Where $p$ is the prior probability density function and $p_0$ represents the base distribution. The Jacobian determinant is positive because we have performed the inverse operation and need to add the logarithmic volume change caused by the transformation to the base log probability.

The conventional KL formula is:

$$D_{\text{KL}}(q(z|x, y) \| p(z|x)) = \mathbb{E}_{q(z|x,y)} \left[ \log q(z|x, y) - \log p(z|x) \right] \tag{10}$$

We insert the log probability of our calculated prior and posterior into the KL formula.

$$\begin{aligned}
\text{KL}(q(z|x, y) \,\|\, p(z|x)) = \mathbb{E}_{z \sim q(z|x,y)} \Bigg[ &\log q_0(z_0|x, y) - \sum_{k=1}^{K_q} \log \left| \det \left( \frac{\partial f_k^{(q)}}{\partial z_{k-1}} \right) \right| \\
&- \log p_0(z_0'|x) - \sum_{k=1}^{K_p} \log \left| \det \left( \frac{\partial f_k^{-1(p)}}{\partial z_{k-1}'} \right) \right| \Bigg]
\end{aligned} \tag{11}$$

Finally, the ELBO of proposed method is as follows:

$$\mathcal{L}_{\text{ELBO}}(x) = \mathbb{E}_{z \sim q(z|x,y)} \left[ \log p(y|x, z) \right] - \beta * \text{KL}\left( q(z|x, y) \,\|\, p(z|x) \right) \tag{12}$$

Where the $\mathbb{E}_{z \sim q(z|x,y)} \left[ \log p(y|x, z) \right]$ is refereed as reconstruction term which we applied the MSE. The weight $\beta$ is used to control the influence of the KL term.

**KL vanishing**: In order to further mitigate the influence of posterior collapse, inspired by various VAE annealing algorithms [18, 15, 4, 20], we set $\beta$ to zero at the initial stage of training, allowing the model to only optimize the reconstruction terms at the initial stage and learn to encode valid information at first. We call this step the Warm-up operation of the model. In the subsequent training, the weight of $\beta$ is gradually increased linearly until the set maximum value of $\beta$ is reached to achieve stability. The annealing algorithm formula of VAE is as follows:

$$
\beta_t = \begin{cases} 0, & \text{if } t < T_1 \text{ (Warm up)}, \\ \dfrac{t - T_1}{T_2} \cdot \beta_{\max}, & \text{if } T_1 \leq t \leq T_1 + T_2 \text{ (Increase)}, \\ \beta_{\max}, & \text{if } t > T_1 + T_2 \text{ (Stable)}. \end{cases} \tag{13}
$$

Where $t$ represents the current training epoch, $T_1$ is the number of warm ups, $T_2$ is the number of epochs from warm up to the stable stage, and $\beta_{\max}$ is the set maximum value of $\beta$.

## 4 Experiment

We evaluated our method on three tasks with varying structural and annotation characteristics: (1) cancer detection with multiple expert annotations capturing boundary uncertainty; (2) crack detection with single annotations focused on fine curvilinear structures; and (3) urban scene segmentation as a multi-label task with single annotations. For fair comparison on the last two deterministic datasets, we evaluated the probabilistic model using single-sample inference to match deterministic baselines.

**Setting**: We evaluate our proposed method by applying it to three representative VAE-based stochastic segmentation models: Probabilistic U-Net, Hierarchical Probabilistic U-Net, and PHISeg. All models are trained using the Adam optimizer for 500 epochs with a batch size of 32 and all experiments are programmed by Pytorch 2.4.1 and conducted using NVIDIA A100 Tensor Core GPU.

**Metrics**: We evaluate performance using a combination of conventional and stochastic segmentation metrics. For general segmentation quality, we report the mean Intersection over Union (mIoU), precision, recall, and F1-score, which provide insight into overall accuracy and class-wise balance. To assess the consistency and diversity of probabilistic predictions, we include the Generalised Energy Distance (GED), a proper scoring rule that evaluates sample quality relative to ground truth distributions, and Hungarian-Matched Intersection over Union (HM-IoU), which measures alignment between sampled predictions and ground truth annotations through optimal bipartite matching.

### 4.1 LIDC-IDRI

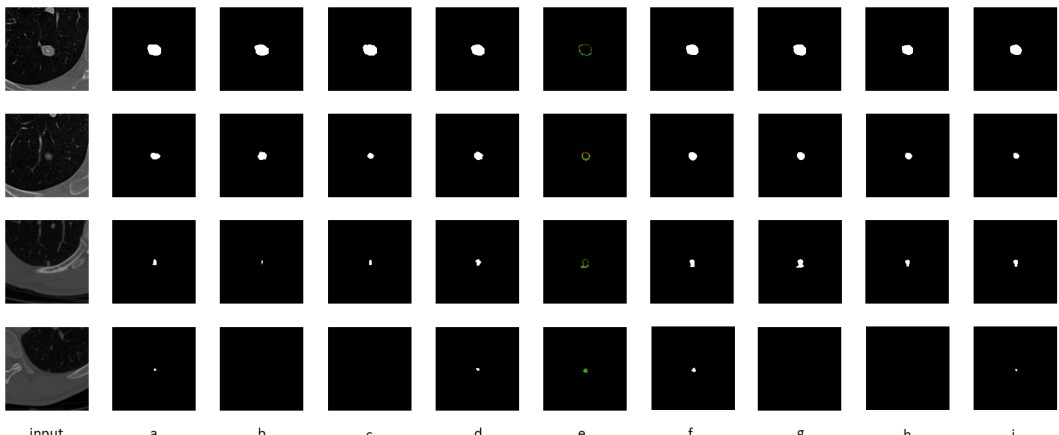

Figure 3: Qualitative results on four LIDC images with proposed method. (a)-(d) represent ground truth label from four different experts. (e) represent the standard deviation of prediction. (f)-(i) represent the prediction from proposed method.

The Lung Image Database Consortium image collection (LIDC-IDRI) consists of thoracic computed tomography (CT) scans for diagnostic and lung cancer screening, with annotated lesions provided by multiple radiologists [2]. Following the experimental setting of CCDM, we extracted a total of 15,096 slices of size 128 × 128, and divided the dataset into training, validation, and testing sets with a ratio of 60:20:20.

Table 1: Quantitative results on LIDC, with methods listed in chronological order. Except for Prob. U-Net + random, all baseline results are taken from [41]. Our proposed method is integrated into three baseline models, each showing substantial performance improvements. Note that lower values of GED indicate better performance, while higher values of HM-IoU are preferred.

| Method | $\text{GED}_{16}$ | $\text{GED}_{32}$ | $\text{GED}_{50}$ | $\text{GED}_{100}$ | $\text{HM-IoU}_{16}$ | $\text{HM-IoU}_{32}$ |
|---|---|---|---|---|---|---|
| Prob. Unet [25] | $0.310_{\pm 0.01}$ | $0.303_{\pm 0.01}$ | – | $0.252_{\pm 0.004}$ | $0.552_{\pm 0.00}$ | $0.548_{\pm 0.00}$ |
| Prob. Unet + random | $0.314_{\pm 0.005}$ | $0.311_{\pm 0.003}$ | – | – | $0.545_{\pm 0.004}$ | $0.548_{\pm 0.005}$ |
| HProb. Unet [24] | $0.270_{\pm 0.01}$ | – | – | – | $0.530_{\pm 0.01}$ | – |
| PhiSeg [3] | $0.262_{\pm 0.00}$ | $0.247_{\pm 0.00}$ | – | $0.224_{\pm 0.004}$ | $0.586_{\pm 0.00}$ | $0.595_{\pm 0.00}$ |
| SSN [28] | $0.259_{\pm 0.00}$ | $0.243_{\pm 0.01}$ | – | $0.225_{\pm 0.002}$ | $0.558_{\pm 0.00}$ | $0.555_{\pm 0.01}$ |
| cFlow [35] | – | $0.225_{\pm 0.01}$ | – | – | – | $0.584_{\pm 0.00}$ |
| CAR [21] | – | – | – | $0.228_{\pm 0.009}$ | – | – |
| JProb. Unet [45] | – | $0.206_{\pm 0.01}$ | – | – | – | $\mathbf{0.647}_{\pm 0.01}$ |
| PixelSeg [44] | $0.243_{\pm 0.01}$ | – | – | – | $0.614_{\pm 0.01}$ | – |
| MoSE [17] | $0.218_{\pm 0.03}$ | $0.195_{\pm 0.002}$ | $0.195_{\pm 0.002}$ | $0.189_{\pm 0.002}$ | $0.624_{\pm 0.004}$ | – |
| AB [6] | $0.213_{\pm 0.01}$ | $0.196_{\pm 0.02}$ | $0.193_{\pm 0.002}$ | – | $0.614_{\pm 0.01}$ | $0.619_{\pm 0.001}$ |
| CIMD [32] | $0.234_{\pm 0.005}$ | $0.218_{\pm 0.005}$ | $0.210_{\pm 0.005}$ | – | $0.587_{\pm 0.01}$ | $0.592_{\pm 0.002}$ |
| CCDM [41] | $0.212_{\pm 0.002}$ | $0.194_{\pm 0.001}$ | $0.187_{\pm 0.002}$ | $\mathbf{0.183}_{\pm 0.002}$ | $0.623_{\pm 0.002}$ | $0.631_{\pm 0.002}$ |
| Prob. Unet + Our | $\mathbf{0.196}_{\pm 0.001}$ | $\mathbf{0.189}_{\pm 0.001}$ | $\mathbf{0.185}_{\pm 0.002}$ | $0.184_{\pm 0.000}$ | $\mathbf{0.641}_{\pm 0.002}$ | $0.644_{\pm 0.001}$ |
| HProb. Unet + Our | $0.224_{\pm 0.002}$ | $0.219_{\pm 0.001}$ | $0.216_{\pm 0.001}$ | $0.215_{\pm 0.001}$ | $0.562_{\pm 0.001}$ | $0.569_{\pm 0.001}$ |
| PhiSeg + Our | $0.250_{\pm 0.000}$ | $0.243_{\pm 0.000}$ | $0.241_{\pm 0.001}$ | $0.237_{\pm 0.001}$ | $0.623_{\pm 0.001}$ | $0.621_{\pm 0.001}$ |

The model performance of baselines is derived from [41]. We used the same experimental data enhancement method as in the paper to rotate the training image by 0 degrees, 90 degrees, 180 degrees and 270 degrees. The models involved in the experiment can be divided into three categories, VAE-based, diffusion-based, and some other different methods such as mixture of expert. Based on the results in the table 1, our best model achieved the best results in 4 of the 6 metrics. Among them, compared with the most popular diffusion-based models, the performance of the proposed method is improved on all metircs. Compared with cFlow using only posterior NF, GMM+NF also achieved better performance on both metrics in probabilistic U-net architectures. JProb.Unet outperforms all other models on $HM - IoU_{32}$, but all three of our models outperformed it. The qualitative analysis results based on probabilistic U-net can be seen in Fig 3. (b)-(e) annotations created for the four experts, (g)-(l) show the sample distribution of the model, and f is the average value after multiple samples.

In addition to the analysis of the proposed method, we also designed an interesting experiment by replacing the latent variable with a random number. The table 1 shows the performance differences between Prob.Unet and Prob.Unet + random. This proves our hypothesis where the conventional multivariate Gaussian distribution is too simple in distribution structure to represent useful information, and its latent variable can only provides some randomness to the model.

## 4.2 Deterministic segmentation

### 4.2.1 Crack500

The Crack500 dataset is designed for pixel-wise pavement crack segmentation and consists of 500 high-resolution images, resulting in 3,368 cropped images of size 360×640 [43]. The dataset is split into 1,897 training, 347 validation, and 1,124 testing images. Building on the setting [9], we employ horizontal flipping, random cropping, and random rotations of 90°, 180°, and 270° as our data augmentation strategies. Additionally, all training samples are cropped to 256 × 256 during training.

For this task, the results of five traditional deep learning baselines are taken from [9]. As shown in the table 3, the performance of the Probabilistic U-Net is significantly lower than that of naive U-net with a notably high recall. This supports our hypothesis: due to the large variance in the latent distribution, the sampled latent variables are overly random, causing the model to produce over-inclusive segmentations. This issue is particularly pronounced in tasks with highly skewed semantics, such as cracks. Our experimental results demonstrate that our method effectively improves prior expressiveness. On the Probabilistic U-Net architecture, our approach is greatly increase in 3 metrics compared with naive Probabilistic U-Net, and achieves state-of-the-art performance. Additionally,

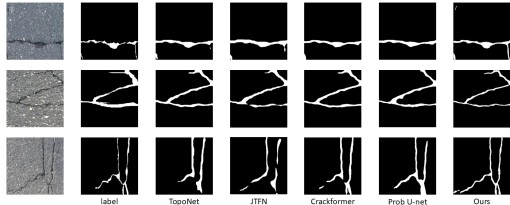

(a) Comparison from Crack500 dataset. All methods are trained and tested by 256 × 256 resolution. It can be known from the image that our segmentation is more compact and close to the boundary

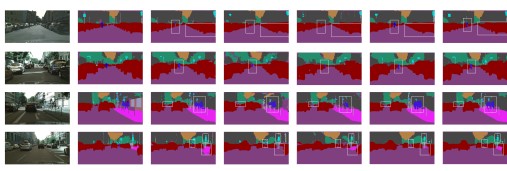

(b) Comparison from Cityscape dataset. All methods are trained and tested by 256 × 512 resolution. The boxes highlights our advantage

Figure 4: Qualitative comparison of deterministic segmentation. All stochastic models (including the proposed method and Prob. U-Net) are evaluated using a single sample. Despite this, our model produces more fine-grained segmentation results compared to mainstream deterministic methods across both tasks.

qualitative results are presented in the Fig 4a, comparing the outputs of baseline models with the improved predictions generated using our proposed method.

Table 2: Quantitative results on Cityscapes. All the results of baselines are from [41]. Param represent the parameter number of model; m represent million

| Method | Backbone | Param | IoU |
|--------|----------|-------|-----|
| DeepLabv3 [5] | ResNet50 | 39m | 58.6 |
| DeepLabv3 [5] | ResNet101 | 58m | 59.2 |
| UPerNet [40] | ResNet101 | 83m | 60.7 |
| HRNet [38] | w48v2 | 70m | 63.3 |
| UPerNet [27] | Swin-Tiny | 58m | 65.5 |
| CCDM [41] | - | 30m | 60.3 |
| CCDM [41] | Dino ViT-S | 50m | 65.8 |
| Prob. Unet [25] | - | 33m | 63.2 |
| Prob. Unet+Our | - | 38m | **73.0** |

Table 3: Quantitative results on Crack500. All the results of baselines are from [9].

| Method | Precision | Recall | F1 |
|--------|-----------|--------|-----|
| UNet [34] | 62.22 | 68.85 | 61.83 |
| VGG-UNet [31] | 58.18 | 60.26 | 51.79 |
| TopoNet [19] | 66.81 | 62.68 | 60.06 |
| DRU [39] | 61.94 | 71.43 | 62.82 |
| Crackformer [26] | 69.13 | 66.24 | 64.75 |
| JTFN [10] | 68.81 | 69.06 | 65.76 |
| JTFN + CIRL [9] | 70.32 | 69.93 | 67.62 |
| Prob. Unet [25] | 56.80 | 69.39 | 60.80 |
| Prob. Unet+Our [9] | **72.97** | **71.72** | **71.80** |

### 4.2.2 Cityscape

Cityscapes is a standard benchmark dataset for multi-class semantic segmentation [11]. It contains 2,975 training images and 500 validation images, each with a resolution of 512 × 1024, annotated across 19 semantic classes.

In this task, the baseline model is taken from CCDM [41].Following proposed setting, we employ resolution 256 × 512 in training and testing set. Our proposed method achieves the best performance within the Probabilistic U-Net architecture. As shown in the table 2, even using a single sample, our model outperforms the latest diffusion-based probabilistic models. Additionally, we provide qualitative results in the Fig 4b, including comparisons with CAVE, diffusion-based models, and a quantitative visualization of our method's segmentation distribution.

### 4.3 Ablation study

In the ablation study, we conducted three different experiments, comparing the performance of conventional Gauss and GMM in the case of NF; Performance comparison with or without NF; The performance of different NF methods is compared. Among the three different tasks, the baseline model has the largest performance variance on the curvilinear task, so we use Crack500 to test the contribution of different modules. F1 score is used to compare the performance of different models in ablation experiments.

Table 4: Ablation study on Crack500 dataset (F1 score). Comparison among different NF types and GMM component numbers.

| Experiment No. | a | b | c | d | e | f |
|---|---|---|---|---|---|---|
| NF Type | NSF | NSF | NSF | - | RealNVP | Glow |
| No. of GMM | 1 | 3 | 6 | 6 | 3 | 3 |
| F1 Score | 66.04 | 71.23 | 71.24 | 61.39 | 69.15 | 69.09 |
| Param | 37.9 | 38.0 | 38.1 | 34.0 | 38.0 | 37.9 |

**Gaussian vs GMM**: We set the number of distributions of the Gaussian mixture model to 1, 3, and 6 respectively, where Gaussian mixture is equivalent to multivariate Gaussian distribution when the number is 1. According to the table 4, we can see that increasing the number of distributions from 1 to 3 in the Gaussian mixture model can bring great performance improvement, while increasing the number from 3 to 6 is relatively less important. This proves that because the Gaussian distribution is too simple, more complex transformations are needed to build a multimodal structure. However, the Gaussian mixture distribution can simulate the data form relatively better, reducing the variation required for NF.

**GMM vs NF**: Next, we test the gain that NF brings to the model. According to the table 4, the performance of the model is significantly improved after NF is used. This proves that the distribution structure close to the data can significantly solve the problems of posterior collapse and insufficient prior expression, and greatly improve the model performance.

**Different NF**: Finally, we experiment the effect of different NF methods on the model performance. These two methods are RealNVP [12] and Glow [22]. Specific explanation of these two NF methods can be found at appendix A.2 and A.3. The results show that the method can better fit the model performance and improve the expression ability of the model.

## 5 Conclusion

In this paper, we propose a novel GMM-based framework to enhance the expressive power of latent space in CVAE-based stochastic segmentation. To validate our hypothesis, we replaced the latent variables with random noise and observed similar results to those using a standard Gaussian prior, indicating that the Gaussian mainly injects randomness rather than meaningful structure. In the processing pipeline, our model incorporates NF in both the prior and posterior networks, yielding a more informative prior compared to existing methods. To stabilize the training, we constrain the ranges of mean and covariance to prevent extreme variance and apply KL annealing to mitigate KL vanishing. The model demonstrates strong performance across three distinct tasks. Notably, we are the first to apply stochastic segmentation to curvilinear structures, addressing the limitations of conventional stochastic models in such scenarios.

However, This study has two main limitations: First, compared to conventional CVAE models, our model introduces a significant increase in computational cost. Although inference speed is not a major concern in target application scenarios of stochastic segmentation, this is still an aspect to improve in our future work. Second, the training process is less stable. Specifically, the normalizing flow requires computing the log-determinant for gradient updates, which makes it more prone to KL vanishing compared to CVAE. As a result, careful tuning of hyperparameters is necessary, along with additional strategies such as annealing schedules and constraining the variance range of GMM components to ensure stable training.

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

# A  Appendix / supplemental material

## A.1  Neural Spline Flows (NSF)

Neural Spline Flows [13] propose to replace the simple element-wise affine transformation in coupling (or autoregressive) layers with more expressive monotonic functions defined by **splines**. Specifically, they use **monotonic rational-quadratic splines**.

### A.1.1  Monotonic Rational-Quadratic Spline Transformation (for a single element $x_i$)

This transformation maps an input $x_i$ to an output $y_i = g(x_i)$. The function $g$ is defined piecewise by $K$ rational-quadratic segments over $K$ bins.

1. **Domain and Knots:** The transformation is typically defined over a bounded interval $[-B, B]$. This interval is partitioned by $K + 1$ knot points $\{(x^{(k)}, y^{(k)})\}_{k=0}^{K}$.

   - Boundary knots: $(x^{(0)}, y^{(0)}) = (-B, -B)$ and $(x^{(K)}, y^{(K)}) = (B, B)$.
   - Internal knots: $K - 1$ internal knot coordinates $(x^{(k)}, y^{(k)})$ for $k = 1, \ldots, K - 1$ are outputs of a neural network (conditioned on the other part of the input, $x_a$, in a coupling layer). These are constrained such that $x^{(0)} < x^{(1)} < \cdots < x^{(K)}$ and $y^{(0)} < y^{(1)} < \cdots < y^{(K)}$ to ensure monotonicity. This is often achieved by parameterizing bin widths and heights via a softmax, then taking cumulative sums.

2. **Derivatives at Knots:** For the spline to be C1 continuous (continuous with continuous first derivative), derivatives $\delta^{(k)} = (dg/dx)|_{x=x^{(k)}}$ are needed at each knot.

   - Boundary derivatives $\delta^{(0)}$ and $\delta^{(K)}$ are often fixed (e.g., to 1, implying linear "tails" outside $[-B, B]$ if $g(x) = x$ there).
   - $K - 1$ internal derivatives $\delta^{(k)} > 0$ for $k = 1, \ldots, K - 1$ are also outputs of the neural network (passed through a softplus function to ensure positivity).

3. **Rational-Quadratic Function in Bin $k$:** Consider the $k$-th bin, from $(x^{(k)}, y^{(k)})$ to $(x^{(k+1)}, y^{(k+1)})$. Let $\xi(x) = \frac{x - x^{(k)}}{x^{(k+1)} - x^{(k)}}$. So, $0 \le \xi \le 1$ as $x$ goes from $x^{(k)}$ to $x^{(k+1)}$. Let $s^{(k)} = \frac{y^{(k+1)} - y^{(k)}}{x^{(k+1)} - x^{(k)}}$ be the slope of the line segment connecting the knots of bin $k$. The transformation $g_k(\xi)$ within this bin is given by (Gregory and Delbourgo, 1982):

$$g_k(\xi) = y^{(k)} + (y^{(k+1)} - y^{(k)}) \cdot \frac{N_k(\xi)}{D_k(\xi)}$$

   where:

$$N_k(\xi) = s^{(k)} \xi^2 + \delta^{(k)} \xi (1 - \xi)$$
$$D_k(\xi) = s^{(k)} + [\delta^{(k+1)} + \delta^{(k)} - 2s^{(k)}] \xi (1 - \xi)$$

   This specific form ensures that $g_k(0) = y^{(k)}$, $g_k(1) = y^{(k+1)}$, $(dg/d\xi)|_{\xi=0} = \delta^{(k)}(x^{(k+1)} - x^{(k)})$, and $(dg/d\xi)|_{\xi=1} = \delta^{(k+1)}(x^{(k+1)} - x^{(k)})$. The constraints on knot positions and positive derivatives ensure $g(x)$ is monotonically increasing.

### A.1.2  Invertibility

To find the inverse $x = g^{-1}(y)$ (or $\xi = g_k^{-1}(y)$ within a bin):

1. First, identify the correct bin $k$ such that $y^{(k)} \le y \le y^{(k+1)}$. This can be done with a binary search since the $y^{(k)}$ values are sorted.

2. Then, solve $y = g_k(\xi)$ for $\xi$. This equation is a quadratic in $\xi$. Let $\Delta y = y - y^{(k)}$ and $\Delta y_{bin} = y^{(k+1)} - y^{(k)}$. The equation becomes:

$$\Delta y \left( s^{(k)} + [\delta^{(k+1)} + \delta^{(k)} - 2s^{(k)}] \xi (1 - \xi) \right) = \Delta y_{bin} \left( s^{(k)} \xi^2 + \delta^{(k)} \xi (1 - \xi) \right)$$

Rearranging terms yields a quadratic equation of the form $a\xi^2 + b\xi + c = 0$:

$$a = \Delta y_{bin} s^{(k)} - \Delta y (\delta^{(k+1)} + \delta^{(k)} - 2s^{(k)})$$

$$b = \Delta y_{bin} \delta^{(k)} + \Delta y (\delta^{(k+1)} + \delta^{(k)} - 2s^{(k)}) - \Delta y s^{(k)}$$

$$c = -\Delta y \delta^{(k)} \quad \text{(Note: Error in previous formula, this term arises from } \xi \text{ coefficient if not careful)}$$

Corrected $c$ : The quadratic derived from $y = g_k(\xi)$ is typically written directly as:

$$(\Delta y_{bin} s^{(k)} - \Delta y (\delta^{(k+1)} + \delta^{(k)} - 2s^{(k)}))\xi^2$$
$$+ (\Delta y_{bin} \delta^{(k)} + \Delta y (\delta^{(k+1)} + \delta^{(k)} - 2s^{(k)}) - \Delta y s^{(k)})\xi$$
$$- \Delta y s^{(k)} = 0 \quad \text{(This form seems more consistent with solving for } \xi \text{ directly)}$$

Since the function is monotonic within the bin, there will be exactly one root $\xi \in [0, 1]$. The solution is typically given as $\xi = \frac{2c}{-b-\sqrt{b^2-4ac}}$ for numerical stability when $4ac$ is small.

3. Once $\xi$ is found, $x = x^{(k)} + \xi(x^{(k+1)} - x^{(k)})$.

The coefficients for the quadratic in $\xi$ to solve are (from Durkan et al., 2019, Appendix A, re-arranged for $a\xi^2 + b\xi + c = 0$):

$$a' = (y^{(k+1)} - y^{(k)})(s^{(k)} - \delta^{(k)}) + (y - y^{(k)})(\delta^{(k+1)} + \delta^{(k)} - 2s^{(k)})$$

$$b' = (y^{(k+1)} - y^{(k)})\delta^{(k)} - (y - y^{(k)})(\delta^{(k+1)} + \delta^{(k)} - 2s^{(k)})$$

$$c' = -s^{(k)}(y - y^{(k)})$$

Then $\xi = \frac{2c'}{-b' - \sqrt{(b')^2 - 4a'c'}}$.

### A.1.3 Jacobian Determinant (Element-wise Derivative)

The derivative $g'(x) = dg/dx$ for $x$ in bin $k$ is:

$$\frac{dg}{dx} = \frac{dg_k}{d\xi} \cdot \frac{d\xi}{dx}$$

We have $\frac{d\xi}{dx} = \frac{1}{x^{(k+1)} - x^{(k)}}$. The derivative $\frac{dg_k}{d\xi}$ is the derivative of the rational-quadratic expression. The paper gives the final form of $g'(x)$ as:

$$g'(x) = \frac{(s^{(k)})^2 \left[\delta^{(k+1)}\xi^2 + 2s^{(k)}\xi(1 - \xi) + \delta^{(k)}(1 - \xi)^2\right]}{D_k(\xi)^2}$$

where $D_k(\xi) = s^{(k)} + [\delta^{(k+1)} + \delta^{(k)} - 2s^{(k)}]\xi(1 - \xi)$. Since $g'(x) > 0$ (due to monotonicity constraints), $\log |g'(x)| = \log(g'(x))$. When used in a coupling layer that transforms $x_b$ based on $x_a$, the total log-determinant is $\sum_i \log(g_i'(x_{b,i}))$, where $g_i'$ is the spline derivative for the $i$-th dimension of $x_b$, parameterized by $x_a$.

### A.1.4 Tails

Outside the interval $[-B, B]$, the transformation is often set to be the identity function ($g(x) = x$) or a linear transformation matching the slope and value of the outermost spline segments at $x = \pm B$, to allow for unbounded inputs.

NSF significantly increases the flexibility of the element-wise transformations compared to simple affine ones, allowing for more expressive models with fewer layers or simpler conditioning networks, while maintaining analytical invertibility and Jacobian computation.

---

### A.2 RealNVP (Real Non-Volume Preserving)

RealNVP [12] introduces **affine coupling layers** designed to have a Jacobian matrix that is triangular, making its determinant easy to compute.

### A.2.1 Affine Coupling Layer Mechanism

Consider an input vector $x \in \mathbb{R}^D$.

1. **Partitioning:** The input $x$ is split into two disjoint parts:
   - $x_a = x_{1:d}$ (the first $d$ dimensions)
   - $x_b = x_{d+1:D}$ (the remaining $D - d$ dimensions)

2. **Transformation:** The output $y \in \mathbb{R}^D$ (with parts $y_a$ and $y_b$) is defined as:

$$y_a = x_a \quad \text{(Identity transformation for the first part)}$$
$$y_b = x_b \odot \exp(s(x_a)) + t(x_a) \quad \text{(Affine transformation for the second part)}$$

Here, $\odot$ denotes element-wise multiplication. The functions $s : \mathbb{R}^d \to \mathbb{R}^{D-d}$ (scale) and $t : \mathbb{R}^d \to \mathbb{R}^{D-d}$ (translation) are typically implemented as neural networks. They take $x_a$ as input and produce the parameters for transforming $x_b$.

### A.2.2 Invertibility

The inverse transformation $x = f^{-1}(y)$ is analytically computable:

1. From $y_a = x_a$, we directly get $x_a = y_a$.

2. Substitute $x_a$ into the second equation to solve for $x_b$:

$$y_b = x_b \odot \exp(s(y_a)) + t(y_a)$$
$$x_b = (y_b - t(y_a)) \odot \exp(-s(y_a))$$

### A.2.3 Jacobian Determinant

The Jacobian matrix of the transformation $f$ (i.e., $\partial y/\partial x$) has the following structure:

$$J = \frac{\partial(y_a, y_b)}{\partial(x_a, x_b)} = \begin{pmatrix} \partial y_a/\partial x_a & \partial y_a/\partial x_b \\ \partial y_b/\partial x_a & \partial y_b/\partial x_b \end{pmatrix}$$

Analyzing the blocks:

- $\partial y_a/\partial x_a = I_d$ (Identity matrix of size $d \times d$)
- $\partial y_a/\partial x_b = \mathbf{0}$ (Matrix of zeros of size $d \times (D - d)$)
- $\partial y_b/\partial x_a$: This block involves derivatives of $s(x_a)$ and $t(x_a)$ with respect to $x_a$. Its exact form is complex but not needed for the determinant.
- $\partial y_b/\partial x_b = \text{diag}(\exp(s(x_a)))$ (A diagonal matrix of size $(D-d) \times (D-d)$ whose diagonal elements are $\exp(s_j(x_a))$ for component $j$)

So, the Jacobian matrix $J$ is:

$$J = \begin{pmatrix} I_d & \mathbf{0} \\ \frac{\partial y_b}{\partial x_a} & \text{diag}(\exp(s(x_a))) \end{pmatrix}$$

This is a lower triangular block matrix. The determinant of a triangular matrix is the product of its diagonal elements.

$$\det(J) = \det(I_d) \cdot \det(\text{diag}(\exp(s(x_a)))) = 1 \cdot \prod_{j=1}^{D-d} \exp(s_j(x_a)) = \exp\left( \sum_{j=1}^{D-d} s_j(x_a) \right)$$

The log-determinant term required for Eq. A1 is $\log|\det(\partial f^{-1}(x)/\partial x)|$. Using the property $\det(A^{-1}) = 1/\det(A)$, we have $\log|\det(\partial f^{-1}(y)/\partial y)| = -\log|\det(\partial f(x)/\partial x)|$. Therefore, for density evaluation with $f^{-1}$, the log-determinant is $\sum_{j=1}^{D-d} s_j(x_a)$, assuming $s(x_a)$ directly outputs the log-scale factors. (If $s(x_a)$ outputs scale factors, then it's $\sum \log|s_j(x_a)|$). The crucial part is that $s(x_a)$ is conditioned on $x_a$, the part of $x$ that is not transformed by this specific affine operation.

### A.2.4 Composition and Alternation

Multiple coupling layers are stacked. To ensure all variables can influence each other, the roles of which part is passed through (identity) and which part is transformed are alternated in successive layers, or a permutation of dimensions is applied between layers.

---

## A.3 Glow

Glow [22] enhances the RealNVP framework by introducing two new operations within each "step" of the flow, in addition to the affine coupling layer. A single step in Glow comprises:

1. Actnorm
2. Invertible 1×1 Convolution
3. Affine Coupling Layer

### A.3.1 Actnorm (Activation Normalization)

This layer performs an element-wise affine transformation per channel:

$$y_{chw} = \alpha_c \cdot x_{chw} + \beta_c$$

where $x_{chw}$ is the activation at channel $c$, height $h$, width $w$. $\alpha_c$ (scale) and $\beta_c$ (bias) are learnable parameters, one per channel $c$.

- **Initialization:** $\alpha_c$ and $\beta_c$ are initialized such that, for the first minibatch of data, the output activations $y_{chw}$ for each channel $c$ have zero mean and unit variance across spatial and batch dimensions. After this initialization, $\alpha_c$ and $\beta_c$ become regular trainable parameters independent of minibatch statistics.
- **Inverse:** $x_{chw} = (y_{chw} - \beta_c)/\alpha_c$
- **Log-Determinant:** The Jacobian is a diagonal matrix. For an input tensor of $H \times W$ spatial dimensions and $C$ channels, the total log-determinant is:

$$\log|\det(J_{\text{actnorm}})| = H \cdot W \cdot \sum_{c=1}^{C} \log|\alpha_c|$$

### A.3.2 Invertible 1×1 Convolution

This operation generalizes fixed permutations. A 1×1 convolution with $C$ input channels and $C$ output channels applies a linear transformation to the channel vector at each spatial location independently. If $x'_{hw} \in \mathbb{R}^C$ is the vector of channels at spatial location $(h, w)$, and $W \in \mathbb{R}^{C \times C}$ is the convolution kernel weight matrix, then:

$$y'_{hw} = W x'_{hw}$$

- **Invertibility:** The matrix $W$ must be invertible. The inverse operation is $x'_{hw} = W^{-1} y'_{hw}$.
- **Log-Determinant:** For an input tensor of $H \times W$ spatial dimensions, the total log-determinant is:
$$\log|\det(J_{\text{1x1conv}})| = H \cdot W \cdot \log|\det(W)|$$
- **Efficient Computation of** $\det(W)$ **and** $W^{-1}$**:** To avoid the $O(C^3)$ cost of directly computing $\det(W)$ and $W^{-1}$, $W$ is parameterized via its **LU decomposition**:
$$W = PLU$$
  where:
  - $P$ is a permutation matrix (fixed, not learned).
  - $L$ is a lower triangular matrix with ones on its diagonal.
  - $U$ is an upper triangular matrix whose diagonal elements $U_{ii}$ are learned (and must be non-zero).

The determinant is then $\det(W) = \det(P) \cdot \det(L) \cdot \det(U) = (\pm 1) \cdot 1 \cdot \prod_{i=1}^{C} U_{ii}$. So, $\log|\det(W)| = \sum_{i=1}^{C} \log|U_{ii}|$. The parameters $L_{ij}$ for $i > j$ and $U_{ij}$ for $i \leq j$ are learned. Inversion $W^{-1} = U^{-1} L^{-1} P^{-1}$ is also efficient due to the triangular nature of $L$ and $U$.

### A.3.3   Affine Coupling Layer

This is the same as in RealNVP.

### A.3.4   Multi-Scale Architecture

Glow also employs a multi-scale architecture. After a few steps of flow at a given resolution:

1. **Squeeze:** Spatial dimensions are reduced and channel dimensions are increased. For $s \times s \times c$ input, it reshapes into $(s/2) \times (s/2) \times 4c$.

2. **Factor Out:** Half of the channels are split off and assumed to follow a Gaussian distribution. These factored-out variables contribute directly to the latent $u$. The other half continue through subsequent flow steps at the new, coarser resolution.

The total log-determinant for a Glow model is the sum of the log-determinants from all actnorm layers, all $1 \times 1$ convolution layers, and all affine coupling layers, plus the log-probability of the factored-out variables under their assumed Gaussian prior.

