# OpenReview forum: "GMM-based VAE model with Normalising Flow for effective stochastic segmentation"
_NeurIPS.cc/2025/Conference — NeurIPS 2025 poster_

### Official Review · Reviewer_kRCB · 2025-06-30

**Clarity:** 4
**Significance:** 4
**Originality:** 4
**Rating:** 5
**Confidence:** 4

**Summary:**

The presented paper introduces a way to modify probabilistic VAEs (specifically tested with Probabilistic UNet, Hierarchical Probabilistic UNet, and PHiSeg) to create a more "expressive" latent space by encoding the latent variables into a Gaussian Mixture Model, followed by using Normalizing Flow with a spline model to transform the variables into a more informative distribution.

**Questions:**

- Are there any other sorts of architectures is this approach suited for integration with besides CVAE?
- How much training time is added by the incorporation of these components over training of the networks alone (as in the LIDC data)?
- Do PHiSeg and Hierarchial Prob UNet, when trained on LIDC, exhibit the same response to randomization of the latent variable as the ProbUNet does?

**Ethical Concerns:**

["NO or VERY MINOR ethics concerns only"]

**Final Justification:**

I thank the authors for their comments and am happy to keep my score at an accept.

**Limitations:**

Limitations are not discussed and should be. Potential negative societal impact is unlikely.

**Quality:**

4

**Strengths And Weaknesses:**

## Quality:
- Strength: the authors' approach improves on state-of-the-field baselines for several disparate tasks, including newer architectures based on diffusion

## Clarity
- Weakness: In figure 3, putting the standard deviation in the middle position leads to confusion - a better structure would be something like "input, stdev, real x4, predicted x4)" , and having labels instead of just a - i

## Significance
- Strength: the improved baselines in many tasks, plus the ability to apply this approach to several other models, mean it is likely to have impact
- Strength: the incorporation of the experiment where the Prob UNet latent variable was randomized was a clever way to look at the issues with this class of models

## Originality
- I am unaware of other work proposing a similar approach but this is the least confident part of my review

---

> ### Author Rebuttal · Authors · 2025-07-29
>
> ### Q: Are there any other sorts of architectures this approach is suited for integration with besides CVAE?
>
> The first thing that comes to our mind is that this combination can be applied in Bayesian Neural Networks (BNN), by replacing the posterior estimation of simple isotropic Gaussian with GMM+NF. Furthermore, according to the comment from reviewer gS3U, this design can also benefit generative semi-supervised learning architectures, such as FlowGMM.
>
> ---
>
> ### Q: Do PHiSeg and Hierarchical Probabilistic U-Net, when trained on LIDC, exhibit the same response to randomization of the latent variable as the ProbUNet does?
>
> According to our experiments, randomizing the latent variables in PHiSeg and Hierarchical Probabilistic U-Net results in a more significant performance drop compared to Probabilistic U-Net. Under the same experimental settings, after applying randomization:
> - For PHiSeg, the $GED_{16}$ score increased from 0.284 to 0.311, and the $HM$-$IoU_{16}$ decreased from 0.570 to 0.524.
> - For Hierarchical Probabilistic U-Net, the $GED_{16}$ score increased from 0.265 to 0.337, and the $HM$-$IoU_{16}$ decreased from 0.555 to 0.518.
>
> We believe this is because both PHiSeg and Hierarchical Probabilistic U-Net adopt a hierarchical modeling strategy. Rather than using a single latent space, they model multiple levels of latent variables. This hierarchical structure distributes the diversity modeling capability across different layers of the network, which offsets the negative effect of sampling from a near-random Gaussian latent space slightly. As a result, we did not observe the same conclusion as reported for Probabilistic U-Net.
>
> **Note:** Since the baseline results in Table 1 of the main paper are directly taken from the CCDM paper, our reproduced results may slightly differ. However, the presented experiment addresses the concern about the effect of latent variable randomization across different models.
>
> ---
>
> ### Q: How much training time is added by the incorporation of these components over training of the networks alone (as in the LIDC data)?
>
> Thank you for your suggestion. In response, we have added additional ablation experiments on the LIDC and Cityscapes datasets in the appendix, including:
>
> 1. The performance of the model with and without the GMM module (when $K=1$)
> 2. The impact of removing the NSF module
> 3. The effect of increasing the number of GMM components $K$
> 4. The effect of increasing the number of NF layers
>
> We also include training time per epoch, peak GPU memory usage, inference latency, and total parameter size for a more comprehensive comparison.
>
> **Ablation Study on Cityscapes**
>
> | Experiment No.          | a     | b     | c     | d     | e     | f     |
> |-------------------------|-------|-------|-------|-------|-------|-------|
> | No. of NSF layer        | 3     | 3     | 3     | -     | 4     | 6     |
> | No. of GMM              | 1     | 3     | 6     | 3     | 3     | 3     |
> | mIOU                    | 68.04 | 71.23 | 71.97 | 65.39 | 69.15 | 69.09 |
> | Train time (s)          | 323   | 355   | 370   | 197   | 569   | 861   |
> | Peak GPU memory (MB)    | 2239.19 | 2239.36 | 2239.87 | 1988.27 | 2240.44 | 2242.48 |
> | Inference latency (s)   | 0.4030 | 0.4129 | 0.4179 | 0.1705 | 0.5894 | 0.7171 |
> | Param size (M)          | 38.0  | 38.0  | 38.1  | 33.7  | 38.0  | 38.0  |
>
> **Ablation Study on LIDC**
>
> | Experiment No.          | a     | b     | c     | d     | e     | f     |
> |-------------------------|-------|-------|-------|-------|-------|-------|
> | No. of NSF layer        | 3     | 3     | 3     | -     | 4     | 6     |
> | No. of GMM              | 1     | 3     | 6     | 3     | 3     | 3     |
> | HM-IoU$_{16}$           | 0.621 | 0.640 | 0.637 | 0.574 | 0.607 | 0.582 |
> | Train time (s)          | 217   | 222   | 245   | 114   | 466   | 739   |
> | Peak GPU memory (MB)    | 738.55 | 738.74 | 738.99 | 687.78 | 739.77 | 741.82 |
> | Inference latency (s)   | 0.3222 | 0.3294 | 0.3311 | 0.0899 | 0.4259 | 0.6114 |
> | Param size (M)          | 37.9  | 38.0  | 38.1  | 33.6  | 38.0  | 38.0  |
>
> As shown in the ablation study in Cityscapes and LIDC, incorporating the normalizing flow module significantly increases inference latency and training time. However, we believe this trade-off is acceptable for the following reasons:  Although NF adds computational overhead, it provides a much more useful latent space, more accurate uncertainty modeling, and more accurate segmentation outputs. These advantages are particularly valuable in high-reliability tasks such as medical image segmentation, where performance and interpretability are crucial. In typical stochastic segmentation scenarios, the focus is often on uncertainty estimation, diversity of predictions, and output reliability rather than on real-time inference speed. Lastly, the current state-of-the-art in stochastic segmentation is diffusion-based models, which require 50 to 1000 denoising steps per sample. For example, CCDM reports an inference latency of 15.67 seconds, significantly higher than ours.
>
> ---
>
> ### Q: Limitations are not discussed and should be
>
> Thank you very much for your reminder. This study has two main limitations:
>
> 1. Compared to conventional CVAE models, our model introduces a significant increase in computational cost. Although inference speed is not a major concern in target application scenarios, it remains a direction for future optimization.
> 2. The training process is less stable due to the use of normalizing flow, which involves computing the log-determinant in gradients. This makes the model more susceptible to KL vanishing than CVAE-based models. Thus, careful hyperparameter tuning and additional stabilizing techniques, such as annealing schedules and variance constraints for the GMM, are necessary.

---

> > ### Comment · Reviewer_kRCB · 2025-08-07
> >
> > Thank you for the additional information, I am happy to keep my score at an Accept.

---

### Official Review · Reviewer_4wT6 · 2025-07-02

**Clarity:** 4
**Significance:** 3
**Originality:** 4
**Rating:** 5
**Confidence:** 3

**Summary:**

This paper addresses the problem of stochastic semantic segmentation by extending the Conditional Variational Autoencoder (CVAE) framework. The authors integrate a Gaussian Mixture Model (GMM) with Normalizing Flows (NF) to model both prior and posterior distributions. A Multiple-Input, Multiple-Output (MIMO) GMM structure is used to induce a structured, multimodal latent space. NF transformations are applied with constraints on means and variances, and a β-annealed KL divergence term is introduced to regulate learning.
The method is evaluated on three tasks: medical nodule segmentation (LIDC-IDRI), curvilinear crack detection (Crack500), and urban scene parsing (Cityscapes). It consistently outperforms probabilistic U-Net and diffusion-based baselines in Generalized Energy Distance (GED), Hungarian-matched IoU, F1-score, and mIoU.

**Questions:**

1. *Statistical robustness across multiple random seeds.*
   Consider reporting mean ± standard deviation over at least three independent runs (using different random seeds for weight initialization and data shuffling), and include paired significance tests (e.g., Wilcoxon signed-rank or permutation tests) when comparing to the probabilistic U-Net on the primary metrics in Tables 1–3.
   *Score impact:* Consistent, significant gains will boost confidence; high variability or non-significance will weaken it.

2. *Quantify computational overhead and memory footprint.*
   Please provide three measures: 1) training time per epoch; 2) inference latency per 3D volume / 2D image, and 3) peak GPU memory usage for the baseline probabilistic U-Net versus your GMM + NF variant on LIDC-IDRI and Cityscapes. Also, please clarify how the number of NF layers and GMM components affects these costs.
   *Score impact:* A clearer description of the method’s efficiency and scalability could positively influence the score, while substantial overhead without justification may raise concerns.

**Ethical Concerns:**

["NO or VERY MINOR ethics concerns only"]

**Final Justification:**

I reviewed the author response, the new multi-seed results, and the added compute/ablation tables. The paper introduces a structured CVAE with a GMM latent and constrained normalizing flows, and shows consistent gains over probabilistic U-Net and diffusion-based baselines on LIDC, Crack500, and Cityscapes. The authors addressed my key concerns by providing (i) three-seed results with mean ± std and (ii) clear reports of training time, inference latency, and peak memory. These additions increase my confidence in the empirical claims and do not change my positive assessment of the technical contribution.

**Limitations:**

The manuscript lacks an explicit discussion of the computational overhead, the robustness to hyperparameters, failure-case examples, and broader risks such as misinterpretation of stochastic masks in clinical or safety-critical settings.
I recommend adding a dedicated “Limitations & Societal Impact” section that quantifies resource usage, analyzes error modes, and reflects on potential negative impacts and dataset biases.

**Paper Formatting Concerns:**

No major formatting issues observed.

**Quality:**

3

**Strengths And Weaknesses:**

*Strengths:*
- Methodological innovation: The method combines a structured GMM latent space with normalizing flows and β-annealed KL. This is a new approach to modeling multimodal uncertainty.
- Broad empirical evaluation: demonstrated gains across medical, structural-defect, and urban-scene datasets, which indicates the generality of the framework.

*Weaknesses:*
- Computational cost is not reported: the additional runtime and memory demands of the GMM + NF pipeline remain unclear, making it hard to assess the practicality of the proposed method.
- Statistical robustness missing: results are presented for a single run per setting, without seed-based variance or significance testing.
- Limited discussion of failure modes: potential risks such as overconfident or spurious segmentations in safety-critical contexts, are not analyzed.

---

> ### Author Rebuttal · Authors · 2025-07-29
>
> ### Q: Statistical robustness across multiple random seeds.
>
> Thank you for the valuable suggestion. We re-evaluated our model using three different random seeds: 194, 683, and 2596. The results are as follows:
>
> 1. On the LIDC dataset, the model achieved $GED_{16}$ to $GED_{100}$ scores of $0.194 \pm 0.002$, $0.191 \pm 0.003$, $0.181 \pm 0.002$, and $0.186 \pm 0.002$, respectively. The $HM\text{-}IoU_{16}$ and $HM\text{-}IoU_{32}$ scores were $0.640 \pm 0.005$ and $0.642 \pm 0.004$, respectively.
> 2. On the Cityscapes dataset, the model achieved an IoU of $72.4 \pm 0.2$.
> 3. On the Crack500 dataset, the precision was $70.15 \pm 0.41$, recall was $71.56 \pm 0.28$, and F1-score was $70.42 \pm 0.27$.
>
> We also evaluated the Probabilistic U-Net under the same settings and conducted a Wilcoxon signed-rank test to compare it with our model. Due to the small sample size ($n=3$), the test yielded a p-value of 0.25, which is the lowest possible resolution at this scale.
>
> ---
>
> ### Q: Quantify computational overhead and memory footprint.
>
> Thank you for your suggestion regarding the ablation study. In response, we have added additional ablation experiments on the LIDC and Cityscapes datasets in the appendix, including:
>
> 1. The performance of the model with and without the GMM module (when $K=1$).
> 2. The impact of removing the NSF module.
> 3. The effect of increasing the number of GMM components $K$.
> 4. The effect of increasing the number of NF layers.
>
> In addition to comparing model performance, we have also included training time per epoch, peak GPU memory usage, inference latency, and total parameter size to provide a more comprehensive analysis.
>
> **Table: Ablation study on Cityscapes dataset**
>
> | Experiment No.         | a     | b     | c     | d     | e     | f     |
> |------------------------|-------|-------|-------|-------|-------|-------|
> | No. of NSF layer       | 3     | 3     | 3     | -     | 4     | 6     |
> | No. of GMM             | 1     | 3     | 6     | 3     | 3     | 3     |
> | mIOU                   | 68.04 | 71.23 | 71.97 | 65.39 | 69.15 | 69.09 |
> | Train time (s)         | 323   | 355   | 370   | 197   | 569   | 861   |
> | Peak GPU memory (MB)   | 2239.19 | 2239.36 | 2239.87 | 1988.27 | 2240.44 | 2242.48 |
> | Infer latency (s)      | 0.4030 | 0.4129 | 0.4179 | 0.1705 | 0.5894 | 0.7171 |
> | Param size (M)         | 38.0  | 38.0  | 38.1  | 33.7  | 38.0  | 38.0  |
>
> **Table: Ablation study on LIDC dataset**
>
> | Experiment No.         | a     | b     | c     | d     | e     | f     |
> |------------------------|-------|-------|-------|-------|-------|-------|
> | No. of NSF layer       | 3     | 3     | 3     | -     | 4     | 6     |
> | No. of GMM             | 1     | 3     | 6     | 3     | 3     | 3     |
> | HM-IoU$_{16}$          | 0.621 | 0.640 | 0.637 | 0.574 | 0.607 | 0.582 |
> | Train time (s)         | 217   | 222   | 245   | 114   | 466   | 739   |
> | Peak GPU memory (MB)   | 738.55 | 738.74 | 738.99 | 687.78 | 739.77 | 741.82 |
> | Infer latency (s)      | 0.3222 | 0.3294 | 0.3311 | 0.0899 | 0.4259 | 0.6114 |
> | Param size (M)         | 37.9  | 38.0  | 38.1  | 33.6  | 38.0  | 38.0  |
>
> As a reference, the peak GPU memory usage of the baseline Probabilistic U-Net is 669.61 MB on LIDC and 1624.30 MB on Cityscapes.
>
> As shown in the ablation study in Cityscapes and LIDC, incorporating the normalizing flow module significantly increases inference latency and training time. However, we believe this trade-off is acceptable for the following reasons:  Although NF adds computational overhead, it provides a much more useful latent space, more accurate uncertainty modeling, and more accurate segmentation outputs. These advantages are particularly valuable in high-reliability tasks such as medical image segmentation, where performance and interpretability are crucial. In typical stochastic segmentation scenarios, the focus is often on uncertainty estimation, diversity of predictions, and output reliability rather than on real-time inference speed. Lastly, the current state-of-the-art in stochastic segmentation is diffusion-based models, which require 50 to 1000 denoising steps per sample. For example, CCDM reports an inference latency of 15.67 seconds, significantly higher than ours.
>
>
> ---
>
> ### Q: Limitations and Social impact
>
> Thank you very much for your reminder. This study has two main limitations:
>
> 1. Compared to conventional CVAE models, our model introduces a significant increase in computational cost. Although inference speed is not a major concern in target application scenarios of stochastic segmentation, this is still an aspect to improve in our future work.
> 2. The training process is less stable. Specifically, the normalizing flow requires computing the log-determinant for gradient updates, which makes it more prone to KL vanishing compared to CVAE. As a result, careful tuning of hyperparameters is necessary, along with additional strategies such as annealing schedules and constraining the variance range of GMM components to ensure stable training.
>
> Failure cases are inevitable in any model. Unlike deterministic segmentation, due to its randomness, stochastic segmentation can determine which regions have high confidence and which have low confidence by outputting multiple possible results and calculating the standard deviation of each pixel point, thereby ensuring that the occasional failed segmentation samples do not cause serious social impact. This is the reason why it can be widely applied to highly sensitive tasks.

---

> > ### Comment · Reviewer_4wT6 · 2025-08-02
> >
> > Thank you for the detailed response and added experiments. These additions increase my confidence in the empirical claims and do not change my positive assessment of the technical contribution.

---

### Official Review · Reviewer_VZXu · 2025-07-02

**Clarity:** 2
**Significance:** 2
**Originality:** 2
**Rating:** 4
**Confidence:** 3

**Summary:**

The paper proposes a novel stochastic segmentation framework by integrating a Gaussian Mixture Model (GMM) with Normalizing Flow (NF) in a Conditional Variational Autoencoder (CVAE), aiming to overcome the limitations of standard Gaussian latent spaces in modeling semantic diversity. GMM helps structure the latent space into distinct semantic clusters, while NF adds flexibility to both prior and posterior distributions, addressing issues like posterior collapse and poor expressiveness. Extensive experiments across LIDC-IDRI, Crack500, and Cityscapes datasets demonstrate state-of-the-art performance in uncertain and fine-grained segmentation tasks.

**Questions:**

Please refer to the Weaknesses.

**Ethical Concerns:**

["NO or VERY MINOR ethics concerns only"]

**Final Justification:**

Thanks for the authors' rebuttal. My major concerns are addressed and I'd like to raise the score to 4.

**Limitations:**

The current version of the paper needs to be improved due to the insufficient evaluation and the misalignment to the motivation.

**Paper Formatting Concerns:**

No significant issues.

**Quality:**

2

**Strengths And Weaknesses:**

Strengths:

1. The combination of GMM and Normalizing Flow can enhance interpretability and expressiveness of the latent space.

2. The model produces competative evaluation results on three datasets.

Weaknesses:

1. The method seems like a simple combination of GMM, VAE, and Normalizing Flow. The authors should provide more insights and theoretical justifications of why we need to integrate all the three components.

2. The motivation and application of this paper is not well-aligned. The paper is motivated to address the uncertainty and annotation variability issues, but only one study case of non-deterministic segmentation task is evaluated. The overall logic is not very convincing: the proposed approach aims to address a stochastic segmentation problem but why does it also outperform deterministic segmentation baselines?

3. Important ablation studies are missing. (1) how do the core components, GMM and NF affect performance? I only see a very simple ablation on Crack500 in the Appendix (which is not mandatory to read for reviewers). (2) How do the components affect stochastic segmentation performance? I believe this comparison is more important than the table the authors show in the Appendix. (3) What happens if we remove GMM?

4. The comparison of computational cost is missing. The new approach introduces additional computations in GMM, which might dominate the whole model. Details of throughput/memory are needd to guarantee fair comparisons.

5. There is another concern regarding the GMM process. If the category count becomes higher, does the number of GMM clusters need to get higher accordingly?

---

> ### Author Rebuttal · Authors · 2025-07-29
>
> ### Q: The method seems like a simple combination of GMM, VAE, and Normalizing Flow. The authors should provide more insights and theoretical justifications of why we need to integrate all the three components.
>
> We appreciate the comment from the reviewer. There may be a misunderstanding between the background and the motivation of our work. Specifically, stochastic segmentation mainly aims to address uncertainty and annotation variability. Many solutions have been proposed for this task, which can be broadly categorized into diffusion-based and CVAE-based approaches.
>
> The main drawback of diffusion-based methods is the lack of interpretability of the distribution. This is critical in security-sensitive tasks such as medical image segmentation, where explainability is essential. On the other hand, CVAE-based methods often suffer from limited data modeling capacity. For instance, in Probabilistic U-Net, replacing the Gaussian distribution with pure random noise still yields comparable performance. This suggests that simple distributions struggle to capture the complexity of real-world data, and thus, the latent variables contribute limited semantic information. In practice, this limitation often manifests as large variances in latent space. There are some attempts to solve these problems, but they have all kinds of drawbacks. Please refer to the related work section for details
>
> Our key innovation lies in increasing the complexity of both the prior and posterior distributions by combining GMM with NF for the first time. This enables better modeling of complex real-world data distributions. The core component is the NF, which transforms a simple base distribution into a more complex one. The GMM is used to provide a more flexible and structured base distribution, thereby helping the NF to learn more effectively and achieve better performance. We adopt the GMM for the following reasons: Compared to a single Gaussian, GMM offers greater representational capacity, allowing the latent space to explicitly encode distinct semantic regions in the image. Through a parameter competition mechanism, each Gaussian component is encouraged to specialize in specific semantic attributes, thereby enhancing both the interpretability of the latent space and the clarity of the resulting segmentations.
>
> ---
>
> ### Q: The motivation and application of this paper is not well-aligned.
>
> Thank you for your comment. The three selected datasets represent three different application scenarios, which allows for a more comprehensive evaluation of our model.
>
> Multi-annotator datasets are relatively scarce. To the best of our knowledge, only LIDC and QUBIQ are publicly available. Among them, LIDC is widely used and validated in the stochastic segmentation community. It provides a fair and consistent benchmark for evaluating model performance and enables direct comparison with previous works which have passed peer reviews. Therefore, we chose LIDC as the multi-annotator dataset for our experiments.
>
> ---
>
> ### Q: The overall logic is not very convincing[...]?
>
> All baseline results in our paper are taken from state-of-the-art publications in their respective areas. Specifically, Table 1 and Table 2 in the main paper are cited from CCDM, a diffusion-based stochastic segmentation model, while Table 3 is cited from JTFN+CIRL.
>
> In fact, many stochastic segmentation papers have evaluated their methods on the Cityscapes dataset and achieved competitive results compared to deterministic models (can check all the baseline methods we compared in table 1 of main paper). It is important to note that if the decoder of a stochastic model does not incorporate latent variables, it essentially becomes a deterministic model. When the latent space is well-modeled, the variance across different outputs is structured and meaningful, rather than being purely random. This indicates that the model’s stochastic behavior is controlled and interpretable.
>
> ---
>
> ### Q: Important ablation studies are missing.
> (1) how do the core components, GMM and NF affect performance?
> (2) How do the components affect stochastic segmentation performance?
> (3) What happens if we remove GMM?
>
> Thank you for your suggestions regarding the ablation experiment section. In fact, these questions have already been addressed in the ablation study on the Crack500 dataset presented in the appendix:
>
> 1. In **Experiment (a)**, we set $K=1$, making the GMM equivalent to a single Gaussian. This allows us to observe the effect of removing GMM from the model.
> 2. In **Experiment (d)**, we remove the NF module and use only the GMM, which helps evaluate the contribution of NF alone.
> 3. **Experiments (b)** and **(c)** show how model performance changes as $K$ increases.
> 4. **Experiments (e)** and **(f)** demonstrate the impact of using different types of NF on the final performance.
>
> In addition, we also added additional ablation experiments on the LIDC and Cityscapes datasets in the appendix, including:
> - The performance of the model with and without the GMM module (when $K=1$).
> - The impact of removing the NSF module.
> - The effect of increasing the number of GMM components $K$.
> - The effect of increasing the number of NF layers.
>
> In addition to comparing model performance, we have also included training time per epoch, peak GPU memory usage, inference latency, and total parameter size to provide a more comprehensive analysis.
>
> **Table: Ablation study on Cityscapes dataset**
> | Experiment No. | a| b| c| d| e| f|
> |----------------|-------|-------|-------|-------|-------|-------|
> | No. of NSF layer | 3   | 3| 3| -| 4| 6|
> | No. of GMM| 1| 3| 6| 3| 3| 3|
> | mIOU| 68.04 | 71.23 | 71.97 | 65.39 | 69.15 | 69.09 |
> | Train time (s)| 323| 355| 370| 197| 569| 861|
> | Peak GPU (MB)| 2239.19 | 2239.36 | 2239.87 | 1988.27 | 2240.44 | 2242.48 |
> | Infer latency (s)| 0.4030 | 0.4129 | 0.4179 | 0.1705 | 0.5894 | 0.7171 |
> | Param size (M)| 38.0|38.0|38.1| 33.7|38.0| 38.0|
>
> **Table: Ablation study on LIDC dataset**
>
> | Experiment No. | a| b| c| d| e| f|
> |----------------|-------|-------|-------|-------|-------|-------|
> | No. of NSF layer | 3| 3| 3| -| 4| 6|
> | No. of GMM| 1   | 3| 6| 3| 3| 3|
> | HM-IOU$_{16}$| 0.621 | 0.640 | 0.637 | 0.574 | 0.607 | 0.582 |
> | Train time (s)| 217| 222| 245| 114| 466| 739|
> | Peak GPU (MB)| 738.55 | 738.74 | 738.99 | 687.78 | 739.77 | 741.82 |
> | Infer latency (s)| 0.3222 | 0.3294 | 0.3311 | 0.0899 | 0.4259 | 0.6114 |
> | Param size (M)| 37.9  | 38.0  | 38.1  | 33.6  | 38.0  | 38.0  |
>
> As shown in the ablation study in Cityscapes and LIDC, incorporating the normalizing flow module significantly increases inference latency and training time. However, we believe this trade-off is acceptable for the following reasons:  Although NF adds computational overhead, it provides a much more useful latent space, more accurate uncertainty modeling, and more accurate segmentation outputs. These advantages are particularly valuable in high-reliability tasks such as medical image segmentation, where performance and interpretability are crucial. In typical stochastic segmentation scenarios, the focus is often on uncertainty estimation, diversity of predictions, and output reliability rather than on real-time inference speed. Lastly, the current state-of-the-art in stochastic segmentation is diffusion-based models, which require 50 to 1000 denoising steps per sample. For example, CCDM reports an inference latency of 15.67 seconds, significantly higher than ours.
>
> ---
>
> ### Q: There is another concern regarding the GMM process. If the category count becomes higher, does the number of GMM clusters need to get higher accordingly?
>
> Thank you for your question; it is indeed a very valuable point. The main purpose of using GMM in our model is to help NF learn better and improve performance. We do not model each semantic label separately; instead, we perform modeling directly on the input image. This implies that the GMM clusters mainly operate in the latent space and are not directly equivalent to the label space. Therefore, the number of GMM components is not inherently related to the number of segmentation labels.
>
> To test this idea, we did a simple experiment on the Cityscapes dataset. We fixed the number of semantic labels to 19 and increased the number of GMM components $K$ step by step. Because we needed to train many models, we only trained each one for 200 epochs to save time.
>
> **Table: Effect of GMM component number $K$ (Cityscapes, 19 labels)**
> | GMM Components ($K$) | mIOU    | Param |
> |----------------------|---------|--------|
> | 1| 68.31| 38.0|
> | 3| 71.72| 38.0|
> | 5| **71.94** | 38.0 |
> | 6| 71.73| 38.1|
> | 7| 71.55| 38.1|
> | 9| 71.62| 38.2|
> | 11| 71.68| 38.2|
> | 13| 71.19| 38.3|
> | 15| 70.26| 38.3|
> | 19| 69.13| 38.4|
>
> However, as shown in the results table, the model achieved the best performance when $K = 5$. After that, the performance fluctuated slightly and then continued to drop when $K > 11$, likely due to too many ineffective clusters. This result indirectly proves our hypothesis. If there were a positive or negative correlation between $K$ and the number of labels, we would expect the performance to steadily increase or decrease as $K$ increases. But instead, the performance first increased, then stayed stable, and finally decreased, which supports our assumption.
>
> Moreover, the ablation study in the appendix also confirms this. On the Crack500 dataset, the performance difference between $K=3$ and $K=6$ is very small.
>
> Due to time and computing limitations, we were only able to conduct a simple experiment and brief analysis on this question. However, we believe the current results provide initial evidence to support our hypothesis. In future work, we plan to perform more detailed experiments to further validate our theory.

---

> ### Comment · Area_Chair_68N2 · 2025-08-08
>
> Dear Reviewer VZXu,
>
> This is the last moment to account for the author feedback and engage in a minimal discussion. Without any discussion your review will weight less than the others.
>
> Best,
>
> Your AC

---

### Official Review · Reviewer_gS3U · 2025-07-03

**Clarity:** 2
**Significance:** 3
**Originality:** 3
**Rating:** 4
**Confidence:** 4

**Summary:**

The paper presents a probabilistic model for stochastic image segmentation based on a Conditional Variational Autoencoder and U-Net framework that integrates Gaussian Mixture Model (GMM) and Normalizing Flow (NF) to improve latent variable representation.
GMM helps create better structure of latent space while NF model image feature transforms to gauge local uncertainty.
The author also introduce some constraints on the range of means and variances to stabilize training of NF.
The proposed approach achieves state-of-the-art results on three different datasets.

**Questions:**

1. How is the single-sample inference conducted? More details should be provided.
2. What are the missing details mentioned in the weaknesses section?

**Ethical Concerns:**

["NO or VERY MINOR ethics concerns only"]

**Final Justification:**

The authors addressed all my concerns so I keep my initial positive rating.

**Limitations:**

No. The authors did not provide limitations of their work.
The authors should address some difficulties in training the model with NF or limitation in the dimension of the latent variables.

**Paper Formatting Concerns:**

I don't find any formatting concerns in this paper.

**Quality:**

3

**Strengths And Weaknesses:**

# Strengths
1. The authors were able to combine two existing components GMM and NF to make them work well together under Probabilistic U-Net framework and improve the overall segmentation results.
2. Experimental results are performed on three datasets for both stochastic and deterministic segmentation and compared with various group of segmentation approaches.
3. Having ablation study to help better understand the role of both GMM and NF contribute to the final results.

# Weaknesses
1. Missing some details on network structures, e.g. posterior and prior networks which were introduced in Probabilistic U-Net. Thus, it's hard to follow without reading prior works.
2. Missing details about the latent space dimension, the number of distributions of the Gaussian mixture model being used in each dataset.
I'm curious to see if there is a correlation between number of semantic classes and the number of distributions.
3. The idea of combining GMM and NF has been done in some prior works, e.g. FlowGMM.

---

> ### Author Rebuttal · Authors · 2025-07-29
>
> ### Q: Missing details about the latent space dimension, the number of distributions of the Gaussian mixture model being used in each dataset.
>
> Thank you very much for your reminder. In our experiments, the number of NSF layers is set to 3 across all three datasets, and the dimensionality of the latent variable is 8. The number of GMM components $K$ is set to 3 for both the LIDC and Crack500 datasets, and 5 for the Cityscapes dataset.
>
> ### Q: I'm curious to see if there is a correlation between number of semantic classes and the number of distributions.
>
> Thank you for your question; it is indeed a very valuable point. The main purpose of using GMM in our model is to help NF learn better and improve performance. We do not model each semantic label separately; instead, we perform modeling directly on the input image. This implies that the GMM clusters mainly operate in the latent space and are not directly equivalent to the label space. Therefore, the number of GMM components is not inherently related to the number of segmentation labels.
>
> To test this idea, we did a simple experiment on the Cityscapes dataset. We fixed the number of semantic labels to 19 and increased the number of GMM components $K$ step by step. If there was a strong link between $K$ and the number of labels, the model should perform better when $K$ gets close to 19. Due to huge data and experiments, limited time and resources, we only trained each one for 200 epochs to save time.
>
> **Table: Effect of GMM component number $K$ (Cityscapes, 19 labels).**
>
> | GMM Components ($K$) | mIOU    | Param |
> |----------------------|---------|--------|
> | 1                    | 68.31   | 38.0   |
> | 3                    | 71.72   | 38.0   |
> | 5                    | **71.94** | 38.0 |
> | 6                    | 71.73   | 38.1   |
> | 7                    | 71.55   | 38.1   |
> | 9                    | 71.62   | 38.2   |
> | 11                   | 71.68   | 38.2   |
> | 13                   | 71.19   | 38.3   |
> | 15                   | 70.26   | 38.3   |
> | 19                   | 69.13   | 38.4   |
>
> However, as shown in the results table, the model achieved the best performance when $K = 5$. After that, the performance fluctuated slightly and then continued to drop when $K > 11$, likely due to too many ineffective clusters. This result indirectly proves our hypothesis. If there were a positive or negative correlation between $K$ and the number of labels, we would expect the performance to steadily increase or decrease as $K$ increases. But instead, the performance first increased, then stayed stable, and finally decreased, which supports our assumption.
>
> Moreover, the ablation study in the appendix also confirms this. On the Crack500 dataset, the performance difference between $K=3$ and $K=6$ is very small.
>
> Due to time and computing limitations, we were only able to conduct a simple experiment and brief analysis on this question. However, we believe the current results provide initial evidence to support our hypothesis. In future work, we plan to perform more detailed experiments to further validate our theory.
>
> ### Q: How is the single-sample inference conducted? More details should be provided.
>
> Our implementation supports two single-sample inference modes. The first is the maximum-confidence mode. In this mode, we first identify the GMM component with the highest posterior mixing coefficient for a given input. We then select the mean of this component $\mu_k$, as the deterministic latent variable for inference. This represents the most likely semantic cluster for the input. The second is the random-sampling mode, where we first sample a component $k$ based on the mixing coefficients, and then sample a latent variable $z$ from that component's Gaussian distribution $\mathcal{N}(\mu_k, \Sigma_k)$. Based on our experiments, there was not a significant difference in performance, so we selected the random-sampling mode in our main experiments to maintain the randomness of our model.
>
> ### Q: The idea of combining GMM and NF has been done in some prior works, e.g. FlowGMM.
>
> Thank you very much for your reminder. Our literature review only explored stochastic segmentation. To the best of our knowledge, we applied GMM+NF for the first time in stochastic segmentation. Papers with similar ideas in other fields have been updated in the literature review.
>
> ### Q: What are the missing details mentioned in the weaknesses section?
>
> Thank you very much for your reminder. This study has two main limitations: First, compared to conventional CVAE models, our model introduces a significant increase in computational cost. Although inference speed is not a major concern in target application scenarios of stochastic segmentation, this is still an aspect to improve in our future work. Second, the training process is less stable. Specifically, the normalizing flow requires computing the log-determinant for gradient updates, which makes it more prone to KL vanishing compared to CVAE. As a result, careful tuning of hyperparameters is necessary, along with additional strategies such as annealing schedules and constraining the variance range of GMM components to ensure stable training.

---

> > ### Comment · Reviewer_gS3U · 2025-08-05
> >
> > I appreciate the detailed response and added experiments, which have increased my confidence in the experimental findings. Therefore, my initial assessment of the technical aspect of the work remains unchanged.
> >
> > The author mentioned the limitation of their method is that it significantly increased the computational cost compared to conventional CVAE models. Can you provide more specific numbers, e.g. 4x, 8x increased or so?

---

> > > ### Author Response · Authors · 2025-08-05
> > > **Response of computational cost**
> > >
> > > Many thanks for your question. I have conducted an additional ablation study based on the other 3 reviewers' questions (particularly for reviewer 4wT6). Here is the effect of different components on time cost and performance:
> > >
> > > We have added additional ablation experiments on the LIDC and Cityscapes datasets in the appendix, including:
> > >
> > > 1. The performance of the model with and without the GMM module (when $K=1$)
> > > 2. The impact of removing the NSF module
> > > 3. The effect of increasing the number of GMM components $K$
> > > 4. The effect of increasing the number of NF layers
> > >
> > > We also include training time per epoch, peak GPU memory usage, inference latency, and total parameter size for a more comprehensive comparison.
> > >
> > > **Ablation Study on Cityscapes**
> > >
> > > | Experiment No.          | a     | b     | c     | d     | e     | f     |
> > > |-------------------------|-------|-------|-------|-------|-------|-------|
> > > | No. of NSF layer        | 3     | 3     | 3     | -     | 4     | 6     |
> > > | No. of GMM              | 1     | 3     | 6     | 3     | 3     | 3     |
> > > | mIOU                    | 68.04 | 71.23 | 71.97 | 65.39 | 69.15 | 69.09 |
> > > | Train time (s)          | 323   | 355   | 370   | 197   | 569   | 861   |
> > > | Peak GPU memory (MB)    | 2239.19 | 2239.36 | 2239.87 | 1988.27 | 2240.44 | 2242.48 |
> > > | Inference latency (s)   | 0.4030 | 0.4129 | 0.4179 | 0.1705 | 0.5894 | 0.7171 |
> > > | Param size (M)          | 38.0  | 38.0  | 38.1  | 33.7  | 38.0  | 38.0  |
> > >
> > > **Ablation Study on LIDC**
> > >
> > > | Experiment No.          | a     | b     | c     | d     | e     | f     |
> > > |-------------------------|-------|-------|-------|-------|-------|-------|
> > > | No. of NSF layer        | 3     | 3     | 3     | -     | 4     | 6     |
> > > | No. of GMM              | 1     | 3     | 6     | 3     | 3     | 3     |
> > > | HM-IoU$_{16}$           | 0.621 | 0.640 | 0.637 | 0.574 | 0.607 | 0.582 |
> > > | Train time (s)          | 217   | 222   | 245   | 114   | 466   | 739   |
> > > | Peak GPU memory (MB)    | 738.55 | 738.74 | 738.99 | 687.78 | 739.77 | 741.82 |
> > > | Inference latency (s)   | 0.3222 | 0.3294 | 0.3311 | 0.0899 | 0.4259 | 0.6114 |
> > > | Param size (M)          | 37.9  | 38.0  | 38.1  | 33.6  | 38.0  | 38.0  |
> > >
> > > As a reference, the peak GPU memory usage, Inference latency, and training time of the baseline Probabilistic U-Net are 669.61 MB, 0.0712s, and 82s on LIDC, and 1624.30 MB, 0.1290s, and 143s on Cityscapes, respectively. Therefore, at inference latency, our model is 3.23x increased compared with Probabilistic U-Net at Cityscape; 4.63x increased at LIDC. At training time, our model is 2.58x increased compared with Probabilistic U-Net at Cityscape; 2.70x increased at LIDC.
> > >
> > > As shown in the ablation studies, incorporating the normalizing flow module significantly increases inference latency and training time. However, we believe this trade-off is acceptable for the following reasons: NF improves the expressiveness of the latent space, resulting in more accurate uncertainty modeling and more structured, diverse segmentation outputs. These improvements are particularly valuable in critical domains like medical imaging, where precision and interpretability are paramount. In stochastic segmentation tasks, inference latency is often a secondary concern compared to the reliability and diversity of the output. Finally, current state-of-the-art diffusion models in stochastic segmentation require 50–1000 denoising steps per inference. Compared to CCDM's inference latency of 15.67 seconds, our approach remains highly competitive.

---

> > > > ### Comment · Area_Chair_68N2 · 2025-08-08
> > > >
> > > > Dear Reviewer gS3U,
> > > >
> > > > Is this feedback satisfactory?
> > > >
> > > > Best,
> > > >
> > > > Your AC

---

> > > > ### Comment · Reviewer_gS3U · 2025-08-08
> > > >
> > > > Thank you for the additional clarification. I am good with the answer from the authors.

---

### Note · Authors · 2025-08-12

Dear Reviewers and Area Chair

We would like to express our sincere gratitude for your time and the insightful feedback you have provided on our work. The review process has been incredibly valuable, and your comments have allowed us to significantly strengthen our paper.

We were particularly encouraged that the reviewers recognized the importance of the problem we are addressing. In response to your suggestions,

We have further verified our work with additional experiments and ablation studies, and all results reinforce our core claims and demonstrate the significant impact of each component in our model. We believe these additions provide a much more thorough validation of our framework and directly address the key questions raised during the review period.

We are confident that our work offers a meaningful contribution to the field and will inspire further research into creating more reliable and interpretable probabilistic models.

Thank you once again for your constructive engagement and for helping us improve our paper.

Sincerely,

---

### Decision · Program_Chairs · 2025-09-17

**Decision:**

Accept (poster)

**Comment:**

a) The authors improve existing probabilistic segmentation models by combining Normalizing Flow (NF) and Gaussian Mixture Models (GMM) in a Conditional Variational Auto Encoder (CVAE) with the aim of improving the latent space expressiveness and capturing the feature uncertainty.

b)
- New and original framework (NF and GMM in CVAE)
- Extensively tested on multiple datasets and SOTA results
- Ablation study to highlight the role of GMM vs NF

c)
- Missing details or references (FlowGMM)
- Missing numerical exp

d) All reviewers are positive about the paper and their reviews mostly overlap. The work is valuable yet it remains incremental as a combination of existing ideas therefore I do not recommend spotlight of oral.

e) The weaknesses cited above were resolved during the rebuttal. All reviewers are satisfy by the authors responses.